**RESEARCH**                                                                                            **Open Access**

# Emphasis on the deep or shallow parts of the tree provides a new characterization of phylogenetic distances

Julia Fukuyama

## Abstract

**Background:** Phylogenetically informed  distances are commonly used in the analysis of microbiome data, and analysts have many options to choose from. Although all phylogenetic distances share the goal of incorporating the phylogenetic relationships among the bacteria, they do so in different ways and give different pictures of the relationships between the bacterial communities.

**Results:** We investigate the properties of two classes of phylogenetically informed distances: the Unifrac family, including weighted, unweighted, and generalized Unifrac, and the DPCoA family, which we introduce here. Through several lines of evidence, including a combination of mathematical, data analytic, and computational methods, we show that a major and heretofore unrecognized cleavage in the phylogenetically informed distances is the relative weights placed on the deep and shallow parts of the phylogeny. Specifically, weighted Unifrac and DPCoA place more emphasis on the deep parts of the phylogeny, while unweighted Unifrac places more emphasis on the shallow parts of the phylogeny. Both the Unifrac and the DPCoA families have tunable parameters that can be shown to control how much emphasis the distances place on the deep or shallow parts of the phylogeny.

**Conclusions:** Our results allow for a more informed choice of distance and give practitioners more insight into the potential differences resulting from different choices of distance.

**Keywords:** Microbiome, Phylogenetic tree, Ordination, Distance

## Background

The sequencing revolution has given us a much more detailed picture of the bacteria that inhabit the world around us. Since the 1990s, biologists have used marker gene studies to investigate the type and number of bacteria anywhere they care to look [1]. In these studies, a gene, assumed to be common to all the bacteria of interest, is amplified by PCR from the total DNA present in the sample and sequenced. In studies of bacterial communities, the marker gene is often the 16S rRNA gene, as it has both conserved regions that can be used to identify it and more variable regions that allow for differentiation between taxa. The resulting sequences are used as operational taxonomic units, and their abundances are used to describe the abundance of the respective taxon

in the community. These marker gene studies represent a considerable advance over previous culture-based methods of characterizing microbial communities because of their ability to identify unculturable bacteria and the much larger number of bacterial taxa they can identify.

However, a major limitation of this type of study is that the sequence of the 16S gene does not necessarily give us the correct assignment of taxa into functional units. In some cases, the sequence of the 16S gene does not give us enough resolution to distinguish between taxa that have very different functions. In other cases, taxa with different 16S sequences can be functionally the same and our analysis would have more power and be more interpretable if we treated them as such. Within the context of a 16S study, nothing can be done to help with a lack of resolution. The opposite problem, of marker gene studies splitting functionally similar taxa into too many independent units, is

Correspondence: jfukuyam@indiana.edu
Department of Statistics, Indiana University, 919 E 10th Street, Bloomington 47408, Indiana, USA

in principle solvable, and in practice, it is dealt with indirectly by using phylogenetically aware methods for data analysis. To this end, several phylogenetically informed distances has been developed, all of which aim to quantify the similarities or dissimilarities among microbial communities. Each one encodes in some way the intuition that communities containing closely related taxa should be considered more similar to each other than communities containing only distantly related taxa, even all of those taxa are technically distinct.

Once the analyst has settled on a definition of distance, he can compute it for each pair of communities in the study, and the distances can then be used for any number of downstream tasks: testing for differences between communities from different environments, clustering communities into groups, looking for gradients in the communities that are associated with other covariates in the study, and so on. The extent to which these methods succeed depends in large part how appropriate the distance is to the underlying biology, and so it is important to understand how exactly the distance measure uses the phylogeny.

In this paper, we shed light on the properties of these distances. We focus in particular on two classes of phylogenetically informed distances: the Unifrac distances and new a set of distances based on double principal coordinates analysis (DPCoA). The Unifrac distances include unweighted Unifrac [2], weighted Unifrac [3], and generalized Unifrac [4]. Weighted and unweighted Unifrac are among the most popular distances for exploratory analysis of microbiome data (e.g., [5–7]) and are often paired together, as for instance in [8, 9]. Generalized Unifrac has also been used in many studies [10–12], more often in the context of association testing than for exploratory analysis. Double principal coordinates analysis comes from the macroecology literature, but both it and distances derived from it have been used to good effect in the analysis of microbiome data [13–16].

Our main result, which we show through a combination of mathematical, data analytic, and computational methods, is that within both classes, there is a gradient in the level at which the phylogeny is incorporated. Weighted Unifrac and DPCoA sit at one end of the gradient and rely more heavily on the deep structure of the phylogeny when compared with unweighted Unifrac and the non-phylogenetic distances, which rely more heavily on the shallow structure in the phylogeny. We can think of weighted Unifrac and DPCoA as agglomerating taxa into large groups or as having only a small number of degrees of freedom, while the distances at the other end of the spectrum do less agglomeration and have more degrees of freedom.

This result is surprising and is backed up by several different lines of evidence. We first show that we can decompose the Unifrac distances by branch in the tree, and that in both real and simulated datasets, weighted Unifrac relies more heavily on the deep branches than unweighted Unifrac. We then show analytically that the unweighted Unifrac distance on using the full phylogenetic tree is equivalent to the distance computed using a "forest" in which many of the connections between the deep branches in the phylogeny have been removed. This result is complemented by computations showing that weighted Unifrac and DPCoA, but not unweighted Unifrac, are insensitive to "glomming" together leaves in the tree.

Before turning to our results, we review the two classes of phylogenetic distances under consideration: the Unifrac distances and the DPCoA distances.

## The Unifrac distances

The Unifrac distances are a group of phylogenetically informed distances, all of which incorporate the phylogenetic structure by considering the abundances of groups of taxa corresponding to the branches of the phylogenetic tree in addition to individual taxon abundances. Here we will consider both unweighted Unifrac [2] and the generalized Unifrac family [4], which includes as a special case weighted Unifrac [3]. More formal definitions are given in the "Methods" section, but for now, let $p_{ib}$ denote the proportion of bacteria in sample $i$ that are descendants of branch $b$.

### Unweighted Unifrac

With this notation, the unweighted Unifrac distance between sample $i$ and sample $j$ is

$$d_u(i,j) = \frac{\sum_{b=1}^{B} l_b |\mathbf{1}(p_{ib} > 0) - \mathbf{1}(p_{jb} > 0)|}{\sum_{b=1}^{B} l_B} \qquad (1)$$

where $l_b$ is the length of branch $b$, $B$ is the number of branches in the tree, and the notation $\mathbf{1}(p_{jb} > 0)$ means the function that evaluates to 1 if $p_{jb} > 0$ and 0 otherwise. Therefore, the term $|\mathbf{1}(p_{ib} > 0) - \mathbf{1}(p_{jb} > 0)|$ in the numerator of (1) describes whether the descendants of branch $b$ are present in only one of the two communities: it is equal to 1 if true and 0 otherwise. We see that the numerator of (1) sums the lengths of the branches which are unique to one of the two communities and the denominator is the sum of the branch lengths, with the result that the entire quantity can be described as the fraction of branches in the tree that are unique to one of the two communities. Note that this quantity depends only on the presence or absence of the taxa, not on their relative abundances.

### Weighted Unifrac

Weighted Unifrac [3] was designed as a variation of unweighted Unifrac that took into account relative abundances instead of relying solely on the presence or absence of each taxon. As with unweighted Unifrac, it can be written in terms of a sum over the branches of the phylogenetic tree.

Using the same notation as before, the raw weighted Unifrac distance between samples $i$ and $j$ is

$$d_w(i,j) = \sum_{b=1}^{B} l_b |p_{ib} - p_{jb}| \tag{2}$$

A normalizing factor can be added to raw weighted Unifrac to account for different areas of the phylogeny being closer to or farther from the root, in which case the distance between samples $i$ and $j$ is defined as

$$d_{wn}(i,j) = \frac{\sum_{b=1}^{B} l_b |p_{ib} - p_{jb}|}{\sum_{b=1}^{B} l_b (p_{ib} + p_{jb})} \tag{3}$$

Although weighted Unifrac was initially described as the sum over branches given above, it was shown in [17] that it can also be written as an earth-mover's distance. If we imagine the bacteria in two samples as piles of earth positioned at their corresponding leaves on the phylogenetic tree, the weighted Unifrac distance between those samples is the minimum amount of work required to move one pile to the other pile.

### Generalized Unifrac

The final category of Unifrac distances we will consider are the generalized Unifrac distances. They were introduced in [4] in an effort to modulate the emphasis placed on more or less abundant lineages and thereby interpolate between unweighted and weighted Unifrac. The generalized Unifrac distance with tuning parameter $\alpha \in [0, 1]$ is defined as follows:

$$d_g(i,j,\alpha) = \frac{\sum_{b=1}^{B} l_b (p_{ib} + p_{jb})^\alpha \left| \frac{p_{ib} - p_{jb}}{p_{ib} + p_{jb}} \right|}{\sum_{b=1}^{B} l_b (p_{ib} + p_{jb})^\alpha} \tag{4}$$

The generalized Unifrac distances do not exactly interpolate between weighted and unweighted Unifrac, but they come close. Generalized Unifrac with $\alpha = 1$ is exactly weighted Unifrac. As $\alpha$ gets closer to 0, the $(p_{ib} + p_{jb})^\alpha$ term serves to upweight branches that have a smaller proportion of descendants. The intuition behind the design was that unweighted Unifrac places more weight on the branches that have lower abundances, and so distances interpolating between the two should have a parameter that allows more or less weight to be placed on the low-abundance branches. Generalized Unifrac with $\alpha = 0$ is not exactly unweighted Unifrac, but it would be if all of the $p_{ib}$ terms were changed to $\mathbf{1}(p_{ib} > 0)$, that is, if we thought of performing generalized Unifrac on a

matrix containing branch descendant indicators intstead of branch descendant proportions.

### Generalized DPCoA distances

The second class of phylogenetically informed distances under consideration are the generalized DPCoA distances. As with the generalized Unifrac distances, the generalized DPCoA distances have a tunable parameter defining a family of distances, and the distances at the endpoints are special cases. For the generalized DPCoA distances, one endpoint is the standard Euclidean distance, which does not incorporate the phylogeny at all, and the other endpoint is the DPCoA distance. We give a brief review of DPCoA and then describe the family of generalized DPCoA distances.

### DPCoA

Double principal coordinates analysis (DPCoA, originally described in [18]) is a method for obtaining low-dimensional representations of species abundance data, taking into account side information about the similarities between the species. For us, the similarity measure is given by the phylogeny, but in principle, it could be anything. To obtain this low-dimensional representation, points corresponding to species are positioned in a high-dimensional space so that the distance between the species points matches the phylogenetic distances between the species. Then, each bacterial community is conceptualized as a cloud of species points weighted by how abundant the species is in that community. Each community is positioned at the center of mass of its cloud of species points, and principal components is used to obtain a low-dimensional representation of the species points.

The procedure is motivated by definitions of $\alpha$ and $\beta$ diversity introduced Rao in [19]: the inertia of the point clouds corresponding to each bacterial community is his measure of $\alpha$ diversity of that community, and the distance between the community points is his measure of $\beta$ diversity. The framework allows for a unified treatment of diversity, with a decomposition of total $\alpha$ diversity into per-site $\alpha$ diversity and between-site $\beta$ diversity, all while taking into account species similarities.

DPCoA was later characterized as a generalized PCA [20], and from that characterization, we can write the distances in the full DPCoA space between communities $i$ and $j$ as

$$d_d(i,j,r) = (\mathbf{x}_i - \mathbf{x}_j)^T \mathbf{Q} (\mathbf{x}_i - \mathbf{x}_j) \tag{5}$$

where $\mathbf{x}_i$ is a vector giving the taxon abundances in sample $i$ and $\mathbf{Q} \in \mathbb{R}^{p \times p}$ is the covariance matrix for a Brownian motion along the tree [21], meaning that $\mathbf{Q}_{ij}$ denotes the length of the ancestral branches common to taxon $i$ and taxon $j$.

### Generalized DPCoA

We turn next to the generalized DPCoA distances. This family of distances was used implicitly in developing adaptive gPCA [22], a phylogenetically-informed ordination method. Here we will define the family explicitly: the generalized DPCoA distance with parameter $r$ is:

$$d_{\mathrm{gd}}(i, j, r) = \tag{6}$$
$$(\mathbf{x}_i - \mathbf{x}_j)^T (r^{-1}\mathbf{I}_p + (1 - r)^{-1}\mathbf{Q}^{-1})^{-1}(\mathbf{x}_i - \mathbf{x}_j)$$

with the same notation as in Eq. (5) and $r \in (0, 1)$.

In adaptive gPCA, the parameter $r$ controls how much prior weight to give to the phylogenetic structure, but we can dispense with that interpretation and simply think of the different values of $r$ as giving us different distances between the samples, just as the parameter $\alpha$ does for generalized Unifrac.

As with the generalized Unifrac distances, the distances given at the endpoints, with $r = 1$ and $r = 0$, help us to understand the family as a whole. In the limit as $r \to 0$, the DPCoA distance reduces to the standard Euclidean distance (the straight-line distance between two points), which has no dependence on the phylogeny. At the other extreme, in the limit as $r \to 1$, the distance reduces to the distance in double principal coordinates analysis [18].

A final technical note: although we defined the DPCoA distances as distances, the initial description was as an inner product, with the distance being derived from that definition. The formulation as an inner product has some useful implications: for example, if we want to use the distances for ordination (to make a low-dimensional representation of the data), we can use generalized PCA instead of multi-dimensional scaling, with the result that the directions in the low-dimensional plot have interpretations in terms of the taxa in the dataset.

### Relationship between Unifrac and DPCoA distances

Although the Unifrac and DPCoA distances have very different derivations, the mathematical representation of the DPCoA distance is quite similar to the mathematical representation of raw weighted Unifrac. As shown in [23], the DPCoA distance can be written as

$$d_{\mathrm{dpcoa}}(i, j) = \left[\sum_{b=1}^{B} l_b \left(p_{ib} - p_{jb}\right)^2\right]^{1/2} \tag{7}$$

This representation of the distances between the community points in DPCoA suggests that DPCoA and weighted Unifrac should give fairly similar descriptions of the relationships between the community points, as the differences between them are analogous to the differences between the $L_1$ and $L_2$ distances. In practice and in the datasets we have investigated, this has held true.

### Non-phylogenetic distances

We will also compare the phylogenetic distances with the Bray-Curtis dissimilarity and the Jaccard index, two non-phylogenetic measures of community similarity commonly used in ecology. Both measures are defined in the "Methods" section, but for the purposes of this paper, it suffices to know that the Bray-Curtis dissimilarity uses information on species abundance, while the Jaccard index uses only the presence or absence of the species at each site.

### Illustrative dataset

We will use data taken from an experiment studying the effects of antibiotic treatment on the human gut microbiome [24] to illustrate the ideas developed in this paper. In the study, fecal samples were taken from three individuals over the course of 10 months, during which time each subject took two 5-day courses of the antibiotic ciprofloxacin separated by six months. Each individual was sampled daily for the 5 days of the antibiotic treatment and the five following days, and weekly or monthly before and after, for a total of 52 to 56 samples per individual. Operational taxonomic units (OTUs) were created using Uclust [25] with 97% sequence identity, and the 16S sequences were aligned to the SILVA reference tree [26], as described previously [24]. All 2582 OTUs were retained for analysis (no abundance filtering was performed). The abundances were transformed using a started log transformation [27], $x \mapsto \log(1 + x)$ as a way of approximately stabilizing the variance [28] and reducing the outsize effect the most abundant OTUs would otherwise have.

## Results

### Weighted Unifrac favors deep branches, unweighted Unifrac favors shallow branches

All of the Unifrac distances can be decomposed by branch of the phylogenetic tree, and we can use this decomposition to investigate deep vs. shallow branch contributions to these distances. The formulas used are given in the "Methods" section, but we give a brief description here.

Recall from Eq. (2) that raw weighted Unifrac is defined as a sum over branches in the tree. Therefore, the contribution of branch $b$ to either raw or normalized weighted Unifrac distance between samples $i$ and $j$ is just the corresponding element in the sum, $l_b|p_{ib} - p_{jb}|$. For generalized Unifrac, the analogous quantity is $l_b(p_{ib} + p_{jb})^{\alpha} \left|\frac{p_{ib} - p_{jb}}{p_{ib} + p_{jb}}\right|$. For unweighted Unifrac, branch $b$ contributes $l_b / \sum_{j=1}^{B} l_B$ if the branch has descendants in both communities, and contributes zero otherwise. We refer to these as the unnormalized branch contributions. Note that the unnormalized branch contribution depends both on the position of the branch in the tree and its length. Since we are interested in understanding the relative importance of

different regions in the tree, and not in branches in themselves, we also normalize by branch length. This involves dividing each of the quantities defined above by $l_b$, giving us the contribution per unit branch length instead of the overall contribution of a branch. From there, we obtain the normalized contribution of each branch over the entire dataset by averaging these contributions over all pairs of samples in the dataset.

Since we are interested in the relative contributions of the deep and shallow branches, we computed cumulative average contributions of the shallowest $p$ fraction of branches, in the tree, for $p$ in a range between .5 and 1. Shallowness is represented by the number of descendants, so the shallowest branches are those with only one descendant, and they correspond to $p = .5$. The deepest branch, at the root, corresponds to $p = 1$. We then plotted these quantities for unweighted Unifrac, weighted Unifrac, and generalized Unifrac with $\alpha = 0, .25, .5$, and $.75$, as shown in Fig. 1.

Looking first at the two extremes, we see that almost 90% of the unweighted Unifrac distance is contributed on average by branches with 9 or fewer descendants (approximately the shallowest 85% of the branches), while only about 25% of the weighted Unifrac distance is contributed by such branches. The deepest 5% of the branches contribute about 50% in weighted Unifrac but almost nothing in unweighted Unifrac. Although it is not possible to read it off of the plot in Fig. 1, a substantial proportion—over 10%—of the weighted Unifrac distance is contributed by branches with 1000 or more descendants, even though there are only 23 such branches out of a total of 5162 total branches in the tree. The generalized Unifrac distances have behavior in between: generalized Unifrac with values of $\alpha$ close to 1 have relatively larger contributions from the deeper branches, and as $\alpha \rightarrow 0$ the deeper branches contribute less and less. Note however that generalized Unifrac with $\alpha = 0$ still puts more weight on the deep branches than unweighted Unifrac. This is consistent

with the definition of generalized Unifrac not exactly interpolating between unweighted and weighted Unifrac.

That the deep branches are more important to weighted Unifrac and the shallow branches more important to unweighted Unifrac is even more apparent when we plot the branch contributions along the tree. We used the same branch contribution computations but this time plotted them along the phylogenetic tree for the two extreme points, unweighted Unifrac and weighted Unifrac. A subtree containing a randomly selected set of 200 leaves and their ancestral branches is shown in Fig. 2. The subtree is shown because the full phylogenetic tree with 2500 leaves is too big to be easily inspected. We see that for weighted Unifrac, the shallow branches (those with few descendants) contribute very little to the distance, and as we move towards the root, the deeper branches contribute larger and larger amounts. Unweighted Unifrac shows the opposite pattern: the shallow branches contribute more to the distance, and the deep branches often contribute nothing at all (the dark purple branches in the left panel of Fig. 2 have zero contribution).

### Weighted Unifrac favors deep branches in simulation experiments

The pattern of unweighted Unifrac relying more heavily on the shallow branches than weighted Unifrac is not specific to the dataset shown in Fig. 1. To investigate the robustness of this finding, we looked at the branch contributions under three simulation strategies. The first two simulations investigate branch contributions in realistic setups, when there is some structure to the communities that is either unrelated to the phylogeny (the first simulation) or related to the phylogeny (the second simulation). In simulation 1, the samples fall into two groups, each of which has its own set of characteristic taxa, and the sets are unrelated to the phylogeny. In simulation 2, the samples fall along a gradient, with the endpoints corresponding to under- or over-representation

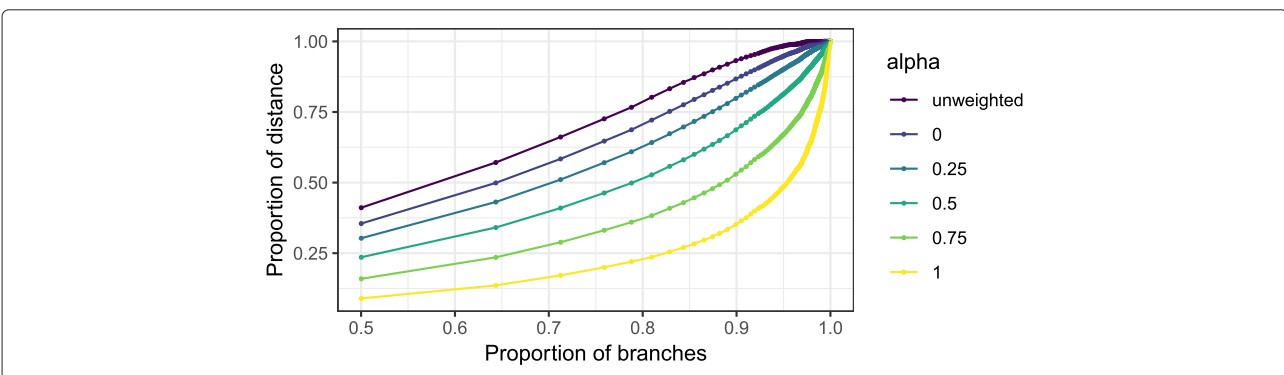

**Fig. 1** Cumulative average contribution (vertical axis) of the shallowest $p$ fraction of the branches in the tree (horizontal axis) to unweighted and generalized Unifrac distances in the antibiotic data. A very large proportion of the unweighted Unifrac distance is contributed by branches with only a few descendants, while that proportion is much smaller for weighted Unifrac

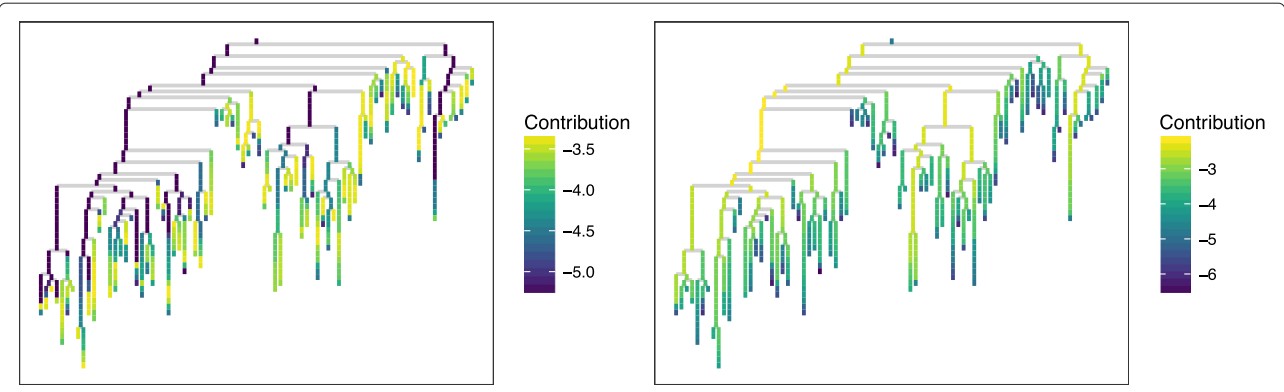

**Fig. 2** Average contributions of each branch to unweighted (left) vs. weighted (right) Unifrac distance. Color represents $\log_{10}$ of the contribution, so numbers closer to zero (more yellow) indicate larger contributions, and large negative numbers (more purple) indicate smaller contributions

of a certain clade. The branch contribution curves are shown in Additional file 1: Figures S1 and S2, and details of the simulation are available in Additional file 1. In each case, for a wide range of numbers of samples, numbers of taxa, numbers of characteristic taxa, and noise in the abundance matrix, we see the same pattern that unweighted Unifrac places more emphasis on the shallow branches than weighted Unifrac does and that the generalized Unifrac distances fall on a spectrum in between.

The last simulation is based on an edge case in which all of the Unifrac distances depend solely on the shallowest branches, those directly above the leaves. The phylogeny is structured as a full binary tree, that is, a tree in which each node has two children, and the tree is taken to have all branches of the same length. The samples are divided into two groups, and for any pair of leaves that share a parent, one leaf is present in the first group and absent in the second, and the other leaf is present in the second group and absent in the first. In this situation, if we have a total of $p$ taxa, the distance between samples in the same group is zero, the unweighted Unifrac distance between samples in different groups is $\frac{p}{2p-2}$, the raw weighted Unifrac distance between samples in different groups is 2, and all of the Unifrac distance, unweighted, weighted, and generalized, is contributed by the branches directly above the leaves. The corresponding branch contribution plot is shown in the upper left panel of Fig. 3. This is the only case we will see where unweighted Unifrac does not place strictly more weight on the shallow branches than weighted Unifrac does, and even so we have equality between the two distances and not a reversal of the pattern.

Next, we looked at what happens to the branch contributions when we add noise to this simulation, as we would see in real data. Instead of letting the taxa we are simulating as being truly present in a sample be deterministically non-zero, we sample counts for those taxa from a double Poisson distribution [29] with a mean of

10 and standard deviations between .01 and 4.5. More details about the simulation strategy and the double Poisson family are given in the "Methods" section, but briefly, the double Poisson is a distribution over the non-negative integers that allows for both under- and over-dispersion relative to the Poisson. When we add even a small amount of noise to the simulation, we immediately recover the pattern of weighted Unifrac placing strictly more weight on the deep branches than unweighted Unifrac, as shown in Fig. 3. As a final note, the amount of noise in panels 2–5 of Fig. 3 is less than we would expect in real experiments. Microbiome counts tend to be overdispersed relative to the Poisson, but the simulations shown in panels 2–5 are substantially under-dispersed. This simulation indicates that even in extreme cases where the Unifrac distances should be determined entirely by the shallowest branches in the tree, when we add any noise to the problem, we recover the pattern of unweighted Unifrac relying more heavily on the shallow branches and weighted Unifrac relying more heavily on the deep branches.

## Unweighted unifrac is independent of the deep structure of the tree

In the previous section, we saw that the deep branches contributed less to the unweighted Unifrac distance than the shallow ones do, and many had zero contribution. Here we strengthen that observation, showing that under conditions that often hold in practice, we can completely remove some of the connections between the deep branches in the tree without changing the set of unweighted Unifrac distances between our samples. This indicates that the set of unweighted Unifrac distances on a given dataset is often completely independent of the deep branching structure of the phylogeny.

Specifically, consider any branch in the tree that has at least one descendant in all of the samples. Note that all the branches ancestral to this branch share the same property. This branch and its ancestors never contribute to

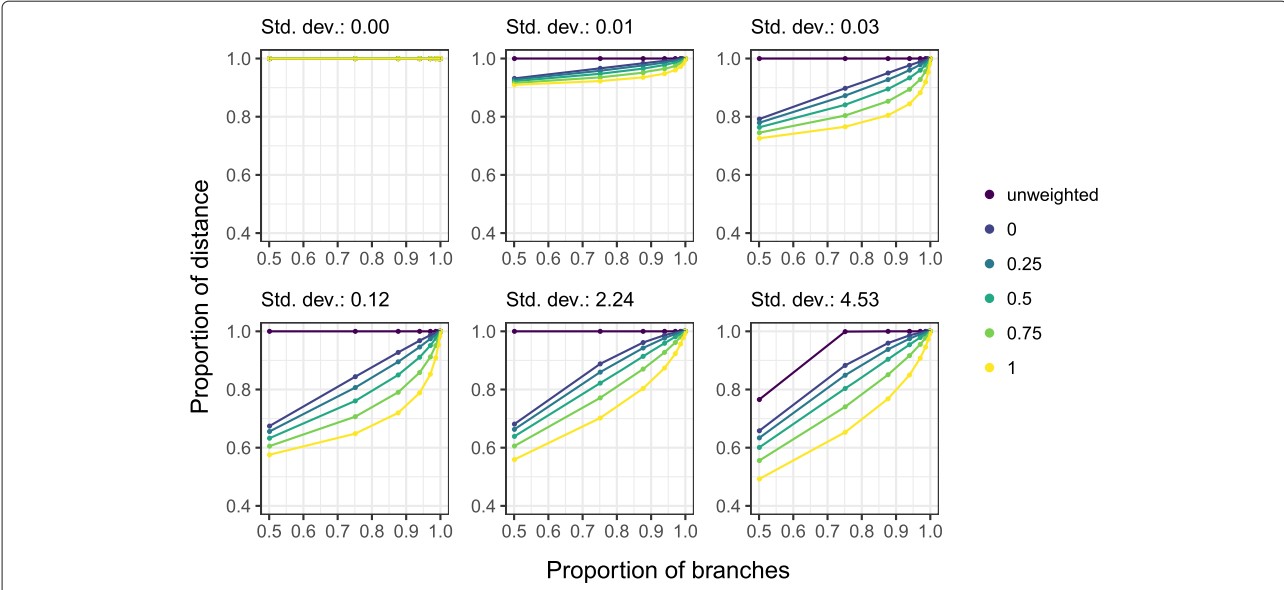

**Fig. 3** Cumulative average contribution (vertical axis) of the shallowest *p* fraction of the branches in the tree (horizontal axis) to unweighted and generalized Unifrac distances for simulated data. Top left panel is the noiseless case, and in subsequent panels, "present" taxa are sampled from a distribution with mean 10 and standard deviation given in the facet label

the unweighted Unifrac distance, and so "breaking" the tree at these branches into unconnected subtrees does not change the set of distances. An illustrative example is shown in Fig. 4, and a more formal proof and description of the equivalence is given in the "Methods" section.

To see how extensively the phylogeny can be broken up and yield the same unweighted Unifrac distances in real data, we performed the procedure of breaking the tree along shared branches on our illustrative dataset. We were interested in the number of subtrees resulting from this procedure and in how many leaves the subtrees contained. In Fig. 5, we see the distribution of the sizes of the 156 resulting trees: out of 2582 taxa, we obtain just under 50 trees with only one leaf. Most of the trees have fewer

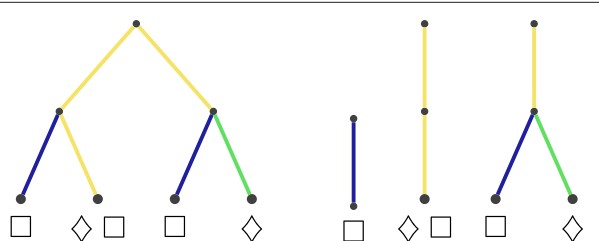

**Fig. 4** Illustration of two sets of trees which give the same unweighted Unifrac distances between a pair of samples. Yellow branches are those with descendants in both communities, and blue or green branches are unique to the square or the diamond communities, respectively. If all the branches have the same length, both the tree on the left and the three-tree forest on the right lead to unweighted Unifrac distances of .5 between the square and diamond communities

than 50 leaves, but we also see some trees with a couple hundred leaves. The large number of small trees is likely responsible for the similarity between the unweighted Unifrac distance and several non-phylogenetic distances, which is explored further in the last part of this section.

## Sensitivity to taxon agglomeration shows that the Unifrac and DPCoA distances are characterized by their reliance on the deep branches

To complement our finding that unweighted Unifrac has no dependence on the deep branching structure, we can show that weighted Unifrac and DPCoA rely primarily on the deep branches by showing that they are relatively insensitive to "glomming" the bacterial taxa together to higher levels on the phylogenetic tree[1]. As with the results for the branch decompositions, we will see that that the generalized Unifrac distances and generalized DPCoA distances show a range of sensitivities to glomming, with DPCoA and weighted Unifrac at the least sensitive end and unweighted Unifrac and the standard Euclidean distance (a non-phylogenetic distance) at the most sensitive end.

When we refer to glomming taxa together here, we mean taking a pair of sister taxa and replacing them with one pseudo-taxon whose abundance is the sum of the abundances of the two taxa which were replaced and whose position on the tree is at the parent node of the two sister taxa. By doing this multiple times, we obtain smaller, lower-resolution datasets with any number of pseudo-taxa between one (all the taxa glommed together into one pseudo-taxon) and the number of taxa in the

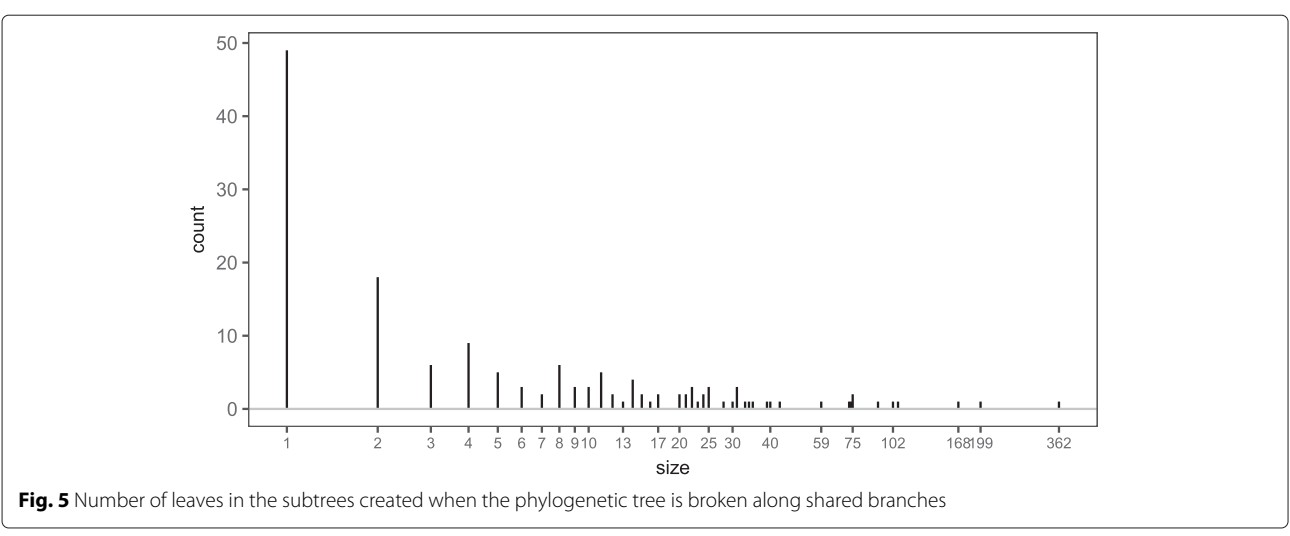

**Fig. 5** Number of leaves in the subtrees created when the phylogenetic tree is broken along shared branches

initial dataset (no glomming). When we glom together taxa, we lose the fine-scale information about the taxon abundances and are left only with information about the abundances of larger clades. If a method gives the same results on heavily glommed data as on the full data, it indicates that the method is not using the fine-scale abundance information.

To quantify the sensitivity of each distance to glomming, we used DISTATIS [30], a method which computes an RV coefficient [31] between distance matrices defined on the same sets of objects. The RV coefficient (described in the "Methods" section) is a generalization to the multidimensional setting of the correlation between vectors, and as for the correlation, higher values indicate that the distances are more similar to each other.

For each distance, we computed the RV coefficient between a dataset glommed to $16, 32, 64, \ldots, 1024$ taxa and the full dataset (with 2582 taxa). These computations were done for members of the Unifrac family, including unweighted Unifrac and generalized Unifrac with $\alpha = 0, .1, .5, .9, 1$, and members of the DPCoA family with values of $r$ between 0 and 1. The results are are shown in Fig. 6, which shows that within each family, there is a range of sensitivity to glomming, with weighted Unifrac (generalized Unifrac with $\alpha = 1$) and standard DPCoA (generalized DPCoA with $r = 1$) being the least sensitive. Within each family, as the tuning parameters decrease, the sensitivity to glomming increases, as we would have expected from our previous results and from the definition of the DPCoA family of distances. DPCoA in particular is quite insensitive to glomming, with the RV coefficient remaining above .98 until we have glommed the initial 2582-taxon tree to under 30 taxa. Weighted Unifrac and some of the generalized Unifrac family members are also relatively insensitive to glomming: a tree an order of magnitude smaller than the full tree still gives

RV coefficients above .95 for all of the generalized Unifrac distances we considered.

The DPCoA distances show more of a range of sensitivities, and by implication in the depth at which they incorporate the phylogeny, than the Unifrac distances do. Standard DPCoA is the least sensitive to glomming out of all the distances under consideration, and the Euclidean distance (generalized DPCoA with $r = 0$) is the most sensitive. That generalized DPCoA with $r = 0$ is the most sensitive to glomming is expected, since it completely ignores the phylogeny. That expectation combined with the result that standard DPCoA is the least sensitive leads us to believe that in general, the DPCoA family of distances will show more of a range in their sensitivity to glomming or the level at which they incorporate the phylogeny than the Unifrac family of distances.

### Comparison of distances to each other shows the same gradient in the Unifrac and DPCoA families

So far, we have seen evidence that within both the Unifrac and DPCoA families, the tunable parameter controls the level at which the phylogeny is incorporated: generalized DPCoA with $r$ close to 1 and generalized Unifrac with $\alpha$ close to 1 both rely heavily on the deep branches of the tree and are remarkably insensitive to glomming together leaves of the phylogeny. On the other end, generalized DPCoA with $r$ close to 0, generalized Unifrac with $\alpha$ close to 0, and unweighted Unifrac have the opposite behavior: they are less dependent on (or in the case of unweighted Unifrac and the standard Euclidean distance, completely independent of) the deep structure in the tree, and they are much more sensitive to glomming together related taxa. The final question we address here is whether the two families follow the same gradient, or whether they give fundamentally different distances between the samples despite exhibiting similar sensitivity to glomming.

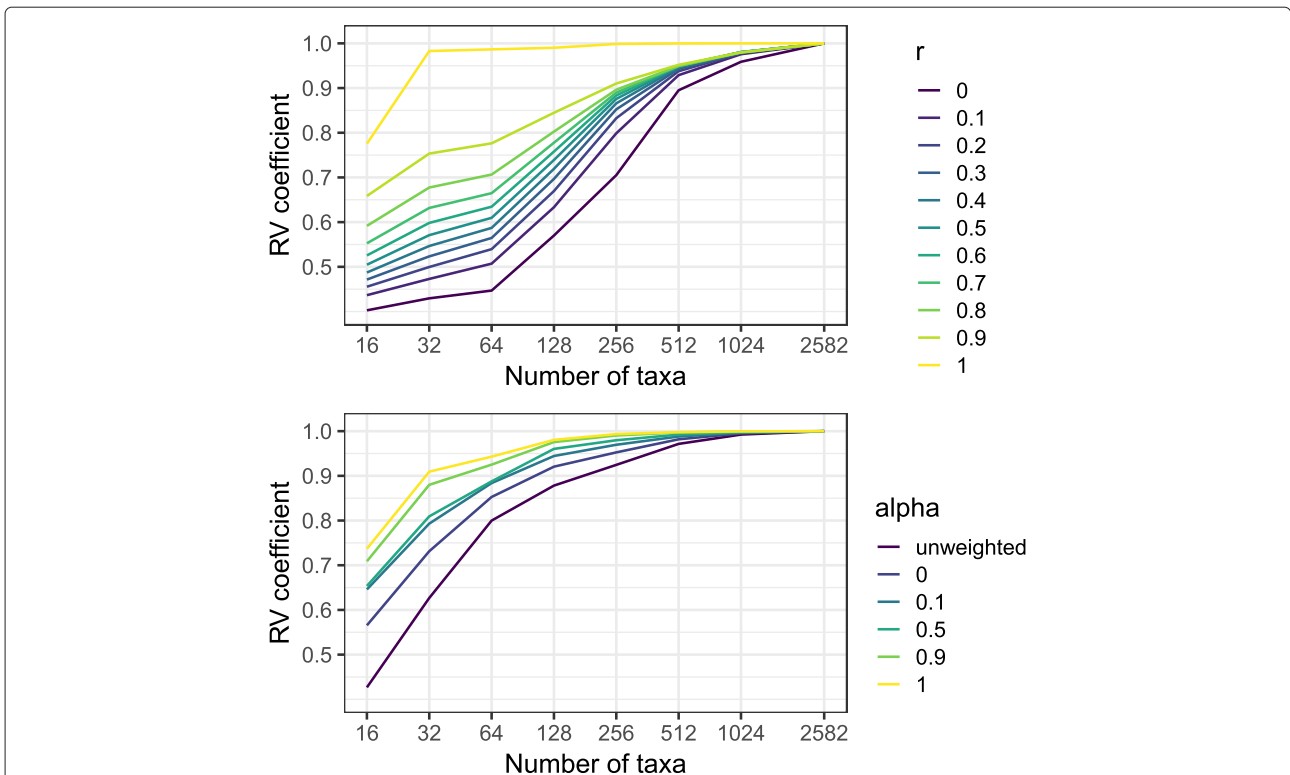

**Fig. 6** The DPCoA and Unifrac distances both exhibit a gradient in their sensitivity to taxon agglomeration. We plot the RV coefficient (vertical axis) between distances computed on the full dataset and distances computed on a dataset glommed to some number of taxa (horizontal axis). We show a set of DPCoA distances (top panel) with different values of *r* (indicated by color) and a set of Unifrac distances (bottom panel) with different values of $\alpha$ (indicated by color)

To this end, we computed generalized Unifrac distances ($\alpha = 0, .1, .25, .5, .9, 1$), the unweighted Unifrac distance, generalized DPCoA distances ($r = 0, .1, \ldots, .9, 1$), the Bray-Curtis dissimilarity ([32]), and the Jaccard dissimilarity ([33]) between the samples in our illustrative dataset. The Bray-Curtis dissimilarity and the Jaccard dissimilarity were included as examples of non-phylogenetic dissimilarities that use either abundance (Bray-Curtis) or solely presence-absence (Jaccard) information about the taxa. We then computed the RV coefficient between each pair of the resulting 20 distances and used DISTATIS to make a low-dimensional visualization of the relationships between the distances.

In Fig. 7, we see that the two families do indeed seem to follow the same gradient. In the representation of the distances along the first two principal axes, we see that the distances corresponding to different values of the tuning parameter ($\alpha$ for generalized Unifrac, *r* for generalized DPCoA) fall along a "horseshoe", within which they are ordered according to the value of $\alpha$ and *r*. We also note that unweighted Unifrac and the non-phylogenetic distances are positioned at the $\alpha = 0/r = 0$ end of the gradient, as we would expect if the gradient is explained by the emphasis the distances place on the deep vs. shallow

branches of the tree. The "horseshoe" phenomenon is a common occurrence in low-dimensional embeddings and is generally considered a mathematical artifact resulting from the projection of a non-linear manifold into a lower-dimensional space (see [34, 35] for mathematical models leading to horseshoes).

We also note that the fraction of variance explained by the first principal axis is over 90%, and the first two principal axes, in which the horseshoe falls, account for more than 96% of the variance explained. This suggests to us that within both families, the differences between the different tuning parameters can be attributed to differences in the level at which the phylogeny is incorporated, and that to a first approximation, the generalized Unifrac and generalized DPCoA families incorporate the phylogeny in the same way.

Although it only accounts for a small fraction, 2.1%, of the explained variance, we also investigated the third principal axis for evidence of either systematic distances between the generalized Unifrac and generalized DPCoA families or between the presence/absence and abundance-based methods (i.e., Jaccard and unweighted Unifrac vs. all the others). In the bottom panel of Fig. 7, we see that the third principal axis separates the generalized Unifrac

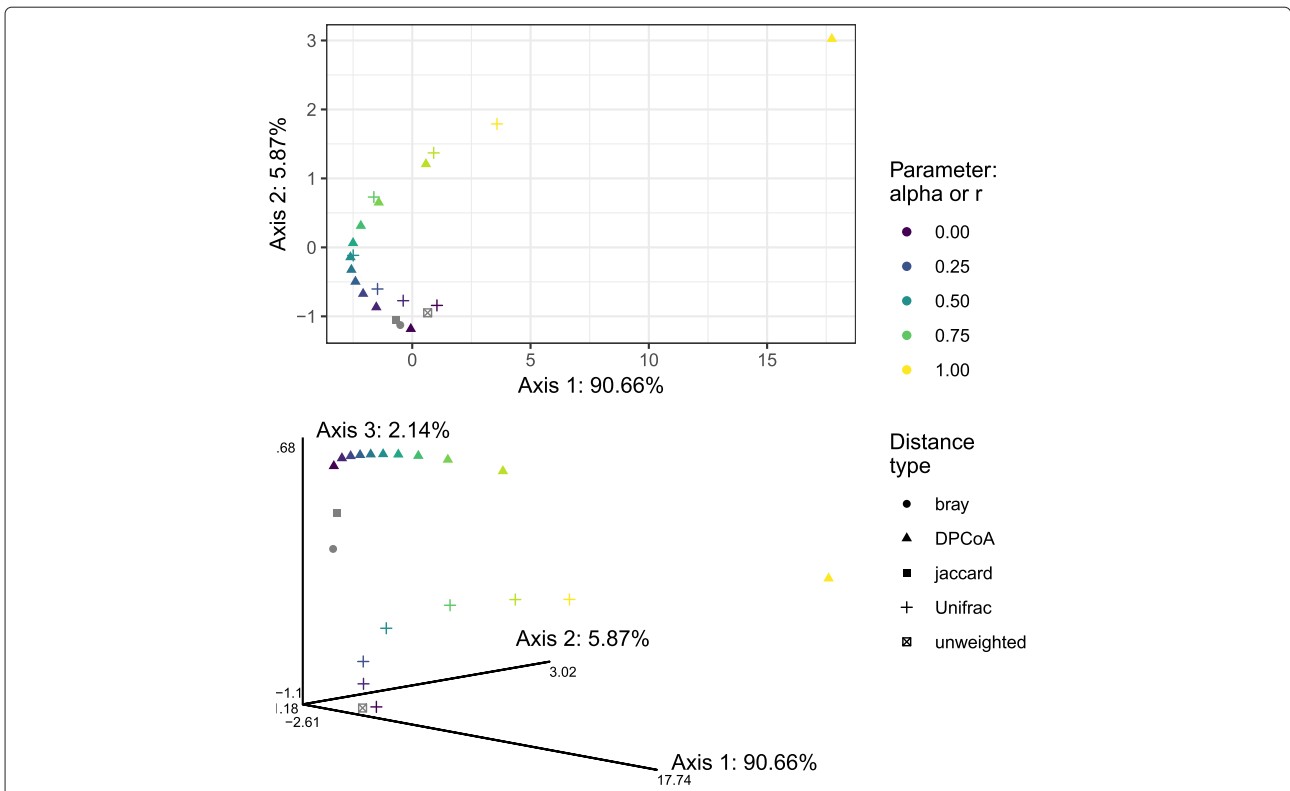

**Fig. 7** DISTATIS representation of the relationships between the generalized Unifrac distances, generalized DPCoA distances, unweighted Unifrac distance, Bray-Curtis dissimilarity, and Jaccard dissimilarity, as computed on the illustrative dataset. Top panel represents the distances on the first two principal axes, bottom panel represents the distances on the top three principal axes

distances from the generalized DPCoA distances and that, furthermore, the separation increases as the value of the tunable parameter decreases and we go towards distances that rely more on the shallow parts of the phylogeny. There is a certain logic to this pattern: distances relying on the deep branches have fewer degrees of freedom, and so there is less room for difference between those distances. The scores on the third axis also fail to separate the presence/absence-based measures and the abundance-based measures: unweighted Unifrac is actually closer to the abundance-based Bray-Curtis measure than it is to the presence/absence-based Jaccard measure, although in the full space the RV coefficients are approximately the same.

## Discussion

Our finding that phylogenetic distances differ in how much they weight different parts of the phylogeny is useful to practitioners who use these distances. The case of unweighted Unifrac compared with weighted Unifrac is especially important, as these two distances are commonly used and often paired together in the same analysis. It is usually assumed that any difference between the two methods is a result of unweighted Unifrac using only presence/absence data and weighted Unifrac using abundance

data, but our results here show that the difference in the emphasis placed on the deep or shallow parts of the phylogeny is perhaps even more important.

Our results are also related to and clarify some previous findings on phylogenetic distances. Parks and Beiko, in [36], catalogued a large number of phylogenetic distances, categorized them according to the set of branches that enter into the mathematical formula for the distances, and examined the empirical similarities between the distances. Their categorization of the distances was as most recent common ancestor (MRCA, distances between two samples depend on only on the most recent common ancestor subtree spanned by the pair of samples), complete lineage (CL, distance is influenced the subtree spanned by the samples and all the branches between that subtree and the root of the tree), and complete tree (CT, the distance is influenced by all of the branches in the tree).

According to this categorization, weighted Unifrac is an MRCA measure, while unweighted Unifrac is a CT measure. This at first seems to be at odds with our results, since a CT measure on a deeper set of branches than an MRCA measure and our results show that in practice, unweighted Unifrac depends more on the shallow branches than weighted Unifrac. However, our results

actually solve something that is a bit puzzling in Parks and Beiko. They find that the categorization of the distances into MRCA/CL/CT does not fit well with the empirical clustering of the distances: the CT classification spans the four clusters they find, and the MRCA and CL classification span three of the four clusters. The results here, both mathematical and empirical, suggest a reason for the lack of alignment: even though unweighted Unifrac technically depends on all of the branches, the form of the distance means that in practice, the deep branches will be less important.

There are of course some limitations to our work. A few of our results are logically entailed by the definitions of the distances, but many will be dataset specific. For instance, branch contributions to unweighted Unifrac *must* be zero for any branch that has descendants in all the samples, but the difference in the fraction of the distance contributed by deep vs. shallow branches and the difference between those contributions for weighted vs. unweighted Unifrac does not have to be as extreme as it is in the dataset we looked at. Additionally, in the datasets we looked at, many of the deep branches could be removed entirely for unweighted Unifrac. We have shown that we can make one break in the tree for every branch that has descendants in all the samples without changing the set of unweighted Unifrac distances. However, this does not mean that in a different dataset we will be able to break the phylogeny up into as many independent pieces as we were able to here.

There is an easy fix for these problems though: simply perform the same calculations on the dataset of interest. If, for example, there is a large difference in the results from unweighted Unifrac vs. weighted Unifrac, the analyst can calculate how much the branches are contributing to the two distances. A big difference in the contributions of the deep vs. shallow branches for the two methods suggests that the difference in results could be due to the difference in how the phylogeny is incorporated.

## Conclusion

We described a new way of characterizing phylogenetic distances, showing that the tunable parameters in both the generalized Unifrac and generalized DPCoA distances control the emphasis placed on the deep vs. shallow branches of the phylogeny. We showed this in several ways: by computing and comparing branch contributions within the Unifrac family, by showing that the families exhibit a gradient in their sensitivity to glomming, and by examining how similar the sets of distances are to each other in real data. In addition to the genereralized Unifrac and generalized DPCoA families, we considered the special case of unweighted Unifrac, showing that that it falls on the end of the spectrum that places more emphasis on the shallow branches of the tree and that it in fact has an

equivalent representation in which the phylogenetic tree is replaced by a "forest" of many independent phylogenies.

Our results give an improved understanding of several phylogenetic distances. This understanding is vital for a valid interpretation of the data and for shaping scientific intuitions about the underlying biology. Our hope is that the properties of these methods that we have outlined will be valuable for the applied researchers who use these tools.

## Methods
### Proof of invariance of unweighted Unifrac to breaking the phylogeny

We first give formal definitions of the tree-related concepts and functions we need to describe manipulations of the phylogenetic tree. We need a definition of a forest to describe how we can break the phylogenetic tree into a forest without changing the unweighted Unifrac distances between the samples.

**Definition 1** *A rooted forest is a triple $F = (V, E, R)$. $V$ is a set of vertices, $E$ is a set of edges on $V$, so that $E \subset \{(v_1, v_2) : v_1, v_2 \in V\}$, and $R \subset V$ is a set of roots. $F$ is such that:*

- *$(V, E)$ is a (possibly disconnected) acyclic graph.*
- *If $V_k$ represents the vertex set of the kth connected component of $(V, E)$, then $R$ is such that $|R \cap V_k| = 1$ for $k = 1, \ldots, K$ (each component has one root).*

The leaf vertices of a forest $F$ are the vertices that only have one neighbor and are not in the root set $R$. The leaf edges of a forest $F$ are the edges that connect to a leaf vertex. The children of a non-leaf vertex $v$ are the vertices that are connected to $v$ by an edge and that are farther from the root. The children of a non-leaf edge $e$ are the edges that share a vertex with $e$ and that are farther from the root.

For notational purposes, we will also assume that the vertex set is $V = \{1, \ldots, |V|\}$ and that if the forest has $p$ leaf vertices they are $\{1, \ldots, p\}$. We further assume that for each edge, if $e = (v_1, v_2)$, $v_1$ closer to the root than $v_2$ implies that $v_1 > v_2$. One way of ensuring these conditions is to use the scheme described in [37].

Unweighted Unifrac requires us to define branch or edge abundances, which we do here with the ndesc function:

**Definition 2** *Let $F = (V, E, R)$ be a rooted forest with $p$ leaf vertices, and let $\mathbf{x} \in \mathbb{N}^p$ represent leaf abundances. The convention that the leaf nodes are $\{1, \ldots, p\}$ and the remaining vertices are $\{p + 1, \ldots, |V|\}$ means that (1) $\mathbf{x}_j$ corresponds to the abundance at leaf vertex j and (2) if edge e is an edge connecting to a leaf node, min(e) will be the leaf node.*

*The ndesc function takes an edge, a leaf abundance vector, and a forest and gives an edge abundance. We define it as:*

$$ndesc(e, \mathbf{x}, F) = \tag{8}$$

$$\begin{cases} \mathbf{x}_{min(e)} & e \text{ a leaf edge} \\ \sum_{e' \in children(e)} ndesc(e', \mathbf{x}, F) & o.w. \end{cases} \tag{9}$$

Note that this definition implies that if $ndesc(e) > 0$, $ndesc(e') > 0$ for any $e'$ ancestral to $e$.

Next, we need a function that describes the tree-breaking operation. The main result will be to show the invariance of the unweighted Unifrac distance to this function under certain conditions.

**Definition 3** *Suppose we have a forest $F = (V, E, R)$ with vertex set $V = 1, \dots, |V|$. Let $e = (v_1, v_2) \in E$.*

*The tree-breaking function tb takes a forest and an edge in the forest and gives a new forest. We define $tb((V, E, R), e) = (V', E', R')$, where*

$$V' = V \cup |V| + 1 \tag{10}$$
$$E' = (E \setminus (v_1, v_2)) \cup (|V| + 1, min(v_1, v_2)) \tag{11}$$
$$R' = R \cup |V| + 1 \tag{12}$$

In words, the edge between $v_1$ and $v_2$ is removed and replaced with a new root node. See Fig. 8 for an illustration, and note that this way of defining the new edge, root, and vertex keeps the vertex assignments consistent with our convention that leaf vertices are labeled $1, \dots, p$ and the remaining vertices are labeled $p + 1, \dots, |V|$.

The following lemma is the main insight into unweighted Unifrac and is fundamentally the reason why

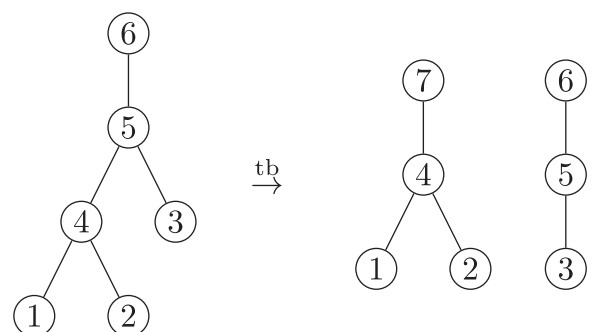

**Fig. 8** Illustration of the tree breaking function. We start off with the six-node tree *T* on the left. If vertex 6 is the root of *T*, its leaves are vertices 1, 2, and 3. When we apply the tree-breaking operation to the (5, 4) edge, we obtain the forest on the right $F = tb(T, (5, 4))$. The roots are now vertices 7 (added when we broke the tree) and 6 (the root in the initial tree) for the two trees in the forest. The leaves remain vertices 1, 2, and 3

we can break the tree in certain ways without changing the unweighted Unifrac distance between the samples.

**Lemma 1** *Let $s(e, F)$ be the sister branch of edge $e$ in forest $F$. If $s(e, F)$ is such that $ndesc(s(e, F), \mathbf{x}, F) > 0$, then*

$$\mathbf{1}(ndesc(e', \mathbf{x}, F) > 0) = \mathbf{1}(ndesc(e', \mathbf{x}, tb(F)) > 0)$$
$$\forall e' \in E(tb(F)) \cap E(F) \tag{13}$$
$$\mathbf{1}(ndesc(e, \mathbf{x}, F) > 0) = \mathbf{1}(ndesc(e'', \mathbf{x}, tb(F)) > 0)$$
$$e'' = E(tb(F)) \setminus E(F) \tag{14}$$

*where $E(F)$ denotes the edge set of forest F.*

*Proof* Consider any edge $e' \in E(F) \cap E(tb(F))$. There are two possibilities: $e$ is a descendant of $e'$ in $F$, or it is not.

– If $e$ is not a descendant of $e'$ in $F$, then

$$ndesc(e, \mathbf{x}, F) = ndesc(e, \mathbf{x}, tb(F)).$$

– If $e$ is a descendant of $e'$ in $F$, then so is $s(e, F)$. In that case, $\mathbf{1}(ndesc(e, \mathbf{x}, F) > 0) = 1$ because $ndesc(s(e, F), \mathbf{x}, F) > 0$. $s(e, F)$ is a descendant of $e'$ in $tb(F)$ as well, and so

$$ndesc(s(e, F), \mathbf{x}, tb(F)) > 0$$

which means that

$$\mathbf{1}(ndesc(s(e, F), \mathbf{x}, tb(F)) > 0) = 1.$$

Therefore, we have (13) for all $e' \in E(tb(F)) \cap E(F)$.

For Eq. (14), let $e''$ be the new edge in $tb(F)$, that is, the sole element of $E(tb(F)) \setminus E(F)$. In that case, $ndesc(e', \mathbf{x}, tb(F)) = ndesc(e, \mathbf{x}, tb(F))$, which implies Eq. (14)

□

In Theorem 1, we use lemma above to show that the tree-breaking function does not change the unweighted Unifrac distance between two samples, denoted $\mathbf{x}_1$ and $\mathbf{x}_2$, if we apply it to the sibling of a branch that has descendants in both samples.

**Theorem 1** *Let $s(e, F)$ denote the sister branch of edge $e$ in forest F. Then, if $s$ is such that $ndesc(\mathbf{x}_1, s, F) > 0$ and $ndesc(\mathbf{x}_2, s, F) > 0$, then $d_u(\mathbf{x}_1, \mathbf{x}_2, F) = d_u(\mathbf{x}_1, \mathbf{x}_2, tb(F, s))$*

*Proof* Our lemma tells us that the tree-breaking function leaves invariant the values of $ndesc(e) > 0$ for every $e \in E \cap E'$, and that $ndesc(e) > 0 = ndesc(e') > 0$ for the comparison between the edge that was removed and the new edge.

□

In Theorem 2, we simply extend Theorem 1 from the unweighted Unifrac distance between a pair of samples to the set of unweighted Unifrac distances between a collection of samples. It describes how we can break the tree and

leave an entire collection of unweighted Unifrac distances among the samples unchanged.

**Theorem 2** *Let* $\mathbf{x}_1, \ldots, \mathbf{x}_n$ *denote leaf abundances for a set of $n$ samples.*

*As before, let $s(e, F)$ denote the sister branch of edge $e$ in forest $F$. If $s$ is such that $\text{ndesc}(\mathbf{x}_i, s, F) > 0$, $i = 1, \ldots, n$, then*

$$d_u(\mathbf{x}_i, \mathbf{x}_j, F) = d_u(\mathbf{x}_i, \mathbf{x}_j, tb(F, s)) \tag{15}$$
$$\forall i = 1, \ldots, n-1, j = i+1, \ldots, n$$

*Proof* This follows by applying Theorem 1 to every pair of samples and noting that our assumption that $s$ has descendants in all the samples implies that $s$ has descendants in every pair of samples.  □

**Branch contributions**

We note that both the weighted and unweighted Unifrac distances are written as a sum over the branches in the tree, and so for any branch, we can ask what fraction of the distance it makes up. Suppose we have a tree or forest $\mathcal{T}$ with $p$ leaves, branches/edges $E$, and an abundance vector $\mathbf{x} \in \mathbb{N}^p$. In the main text, we described quantities $p_{ib}$ as the proportion of bacteria in sample $i$ that are descendants of branch $b$. With the notation in the previous section, we can make the definition

$$p(b, \mathbf{x}, \mathcal{T}) = \frac{\text{ndesc}(b, \mathbf{x}, \mathcal{T})}{\sum_{j=1}^{p} \mathbf{x}_j}, \tag{16}$$

and so if $\mathbf{x}_i$ is the vector containing the abundances of sample $i$, the $p_{ib}$ in, e.g., Eqs. (1), (2), (3), (4), and (7) in the main text would be $p(b, \mathbf{x}_i, \mathcal{T})$.

If we have communities $\mathbf{x}_1$ and $\mathbf{x}_2$ related by a tree or forest $T$ with $B$ edges, the unweighted Unifrac distance between $\mathbf{x}_1$ and $\mathbf{x}_2$ is

$$d_u(\mathbf{x}_1, \mathbf{x}_2, \mathcal{T}) =$$
$$\sum_{b=1}^{B} l_b \frac{|\mathbf{1}(p(b, \mathbf{x}_1, \mathcal{T}) > 0) - \mathbf{1}(p(b, \mathbf{x}_2, \mathcal{T}) > 0)|}{\sum_{b=j}^{B} l_j} \tag{17}$$

and the proportion of the unweighted Unifrac distance contributed by branch $b$ will be

$$\text{ufcont}(b, \mathbf{x}_1, \mathbf{x}_2, \mathcal{T}) =$$
$$l_b \frac{|\mathbf{1}(p(b, \mathbf{x}_1, \mathcal{T}) > 0) - \mathbf{1}(p(b, \mathbf{x}_2, \mathcal{T}) > 0)|}{(\sum_{b=j}^{B} l_j) d_u(\mathbf{x}_1, \mathbf{x}_2, \mathcal{T})} \tag{18}$$

where $l_b$ denotes the length of edge $b$.

The raw weighted Unifrac distance between $\mathbf{x}_1$ and $\mathbf{x}_2$ will be

$$d_w(\mathbf{x}_1, \mathbf{x}_2, \mathcal{T}) = \sum_{b=1}^{B} l_b \left| p(b, \mathbf{x}_1, \mathcal{T}) - p(b, \mathbf{x}_2, \mathcal{T}) \right| \tag{19}$$

the proportion of the raw weighted Unifrac distance contributed by branch $b$ will be

$$\text{wufcont}(b, \mathbf{x}_1, \mathbf{x}_2, \mathcal{T}) =$$
$$l_b \left| p(b, \mathbf{x}_1, \mathcal{T}) - p(b, \mathbf{x}_2, \mathcal{T}) \right| / d_w(\mathbf{x}_1, \mathbf{x}_2, \mathcal{T}) \tag{20}$$

Finally, the generalized Unifrac distance with parameter $\alpha$ between $\mathbf{x}_1$ and $\mathbf{x}_2$ is

$$d_g(\mathbf{x}_1, \mathbf{x}_2, \alpha, \mathcal{T}) =$$
$$\sum_{b=1}^{B} \left( l_b \left[ p(b, \mathbf{x}_1, \mathcal{T}) + p(b, \mathbf{x}_2, \mathcal{T}) \right]^{\alpha} \right.$$
$$\left. \times \left| \frac{p(b, \mathbf{x}_1, \mathcal{T}) - p(b, \mathbf{x}_2, \mathcal{T})}{p(b, \mathbf{x}_1, \mathcal{T}) + p(b, \mathbf{x}_2, \mathcal{T})} \right| \right) \tag{21}$$

and the proportion of the generalized Unifrac distance contributed by branch $b$ is

$$\text{gufcont}(b, \mathbf{x}_1, \mathbf{x}_2, \alpha, \mathcal{T}) =$$
$$l_b \left[ p(b, \mathbf{x}_1, \mathcal{T}) + p(b, \mathbf{x}_2, \mathcal{T}) \right]^{\alpha}$$
$$\times \left| \frac{p(b, \mathbf{x}_1, \mathcal{T}) - p(b, \mathbf{x}_2, \mathcal{T})}{p(b, \mathbf{x}_1, \mathcal{T}) + p(b, \mathbf{x}_2, \mathcal{T})} \right| / d_{guf}(\mathbf{x}_1, \mathbf{x}_2, \alpha, \mathcal{T}) \tag{22}$$

To account for the fact that the different branches have different lengths, we can define the proportion of the distance per unit branch length, which will be the quantities in (18), (20), and (22) divided by $l_b$.

With these definitions, we can find how much on average each branch contributes to the distance. Given a set of community points and a branch in the tree, we can find how much the branch contributes to the distance between every pair of community points. Doing this for every branch gives us an idea of how much of the overall distance is contributed by each of the branches. Suppose that we have a dataset with $n$ communities whose abundances are given in the vectors $\mathbf{x}_1, \ldots, \mathbf{x}_n$. Then, the average contribution of the $b$th branch to the unweighted Unifrac distance, normalized by branch length, is

$$\frac{2}{n(n+1)} \sum_{i=1}^{n-1} \sum_{j=i+1}^{n} \text{ufcont}(b, \mathbf{x}_i, \mathbf{x}_j, \mathcal{T}) / l_b. \tag{23}$$

For generalized Unifrac with parameter $\alpha$, we use the analogous expression:

$$\frac{2}{n(n+1)} \sum_{i=1}^{n-1} \sum_{j=i+1}^{n} \text{gufcont}(b, \mathbf{x}_i, \mathbf{x}_j, \alpha, \mathcal{T}) / l_b. \tag{24}$$

## RV coefficient

The RV coefficient is a generalization of the standard correlation coefficient from vectors to matrices, and was first described in [31]. Suppose that $\mathbf{X} \in \mathbb{R}^{n \times p}$ and $\mathbf{Y} \in \mathbb{R}^{n \times q}$ are two sets of measurements on the same objects, and let $\mathbf{S}_{xx} = \mathbf{X}^T\mathbf{X}$, $\mathbf{S}_{xy} = \mathbf{X}^T\mathbf{Y}$, $\mathbf{S}_{yx} = \mathbf{Y}^T\mathbf{X}$, and $\mathbf{S}_{yy} = \mathbf{Y}^T\mathbf{Y}$. Then the RV coefficient between $\mathbf{X}$ and $\mathbf{Y}$ is defined as

$$RV(\mathbf{X}, \mathbf{Y}) = \frac{\text{tr}(\mathbf{S}_{xy}\mathbf{S}_{yx})}{\sqrt{\text{tr}(\mathbf{S}_{xx})^2 \text{tr}(\mathbf{S}_{yy})^2}} \tag{25}$$

If $p = q = 1$ and $\mathbf{X}$ and $\mathbf{Y}$ are both centered, it is easy to see that the expression above is the square of the standard correlation coefficient $\rho(\mathbf{x}, \mathbf{y}) = \frac{\text{cov}(\mathbf{x},\mathbf{y})}{\sqrt{\text{var}(\mathbf{x})\text{var}(\mathbf{y})}}$.

## Non-phylogenetic distances

For completeness, we give definitions of the Bray-Curtis dissimilarity and the Jaccard index here.

### Bray-Curtis

The Bray-Curtis dissimilarity [32] aims to describe the compositional differences between pairs of communities, and if $\mathbf{x}_1$ and $\mathbf{x}_2$ are vectors describing the species abundances in two communities, the Bray-Curtis dissimilarity between them is defined as

$$d_{\text{BC}}(\mathbf{x}_1, \mathbf{x}_2) = \frac{\sum_{j=1}^{p} |\mathbf{x}_{1j} - \mathbf{x}_{2j}|}{\sum_{j=1}^{p} \mathbf{x}_{1j} + \sum_{j=1}^{p} \mathbf{x}_{2j}} \tag{26}$$

### Jaccard

The Jaccard index [33] is based on the presence or absence of species in each of the communities. If we let $A$ be the set of species present in one community and $B$ be the set of species present in the other, then the Jaccard index is $|A \cap B|/|A \cup B|$. This is commonly transformed into a dissimilarity measure by taking the complement, or

$$d_{\text{jacc}} = 1 - \frac{|A \cap B|}{|A \cup B|} \tag{27}$$

which is what we will use. The Jaccard index is 1 or the Jaccard dissimilarity is 0 when the two communities have the same set of species, and the Jaccard index is 0 or the Jaccard dissimilarity is 1 when the two communities have completely disjoint sets of species.

## Simulation setup

Simulation 3 investigated the case where all of the contributions to the Unifrac distances come from the shallowest branches if the abundances are measured without noise. The simulated datasets contained $p = 512$ taxa and $n = 100$ samples. The phylogenetic tree describing the relationships among the species was a full binary tree, that is, one in which every interior node has two descendants. We let the taxa be numbered $1, 2 \ldots, 512$ and assign them to the leaves of the tree so that pairs of taxa of the form $(2i - 1, 2i)$ for $i = 1, \ldots, 256$ are sister taxa. The mean matrix $M \in \mathbb{R}^{n \times p}$ is then given by

$$M_{ij} = \begin{cases} 10 & i \leq 50, j \text{ is even} \\ 10 & i > 50, j \text{ is odd} \\ 0 & \text{o.w.} \end{cases}$$

Taxon abundance matrices $X \in \mathbb{R}^{n \times p}$ were generated as $X_{ij} \sim$ Double Poisson$(M_{ij}, s)$, using the `rdoublepoisson` function in the `rmutil` package in R [38].

The notation Double Poisson$(m, s)$ indicates a double Poisson distribution with mean $m$ and dispersion parameter $s$. The double Poisson distribution [29] has probability mass function

$$p(y) = c(m,s)s^{y/m}\left(\frac{m}{y}\right)^{y\log s}\frac{y^{y-1}}{y!}$$

where $c(m, s)$ is a normalizing constant, $m$ is the mean parameter, and $s$ is the dispersion parameter. The simulation results shown in Fig. 3 correspond to $s \in \{200, 150, 100, 2, .5\}$. The mean and variance of the double Poisson with mean $m$ and dispersion $s$ are approximately $m$ and $m/s$, respectively, but the standard deviations on the plots were computed by Monte Carlo, as the approximation of the variance as $m/s$ breaks down for the very large values of $s$ used in the simulation.

## Endnote

[1] For another example of glomming in the context of the Unifrac distances, see [39], where glomming was used to cut computation time.

## Additional files

**Additional file 1:** Supplemental methods and figures. Detailed description of simulations 1 and 2, Supplemental Figure S1, and Supplemental Figure S2. (PDF 78 kb)

**Additional file 2:** Review history. (PDF 95 kb)

**Acknowledgements**
JF would like to thank Susan Holmes, David Relman, and Les Dethlefsen for conversations that lay the groundwork for the ideas in this paper, as well as Amy Willis and an anonymous reviewer for their thoughtful comments on the manuscript.

**Authors' contributions**
JF designed the study, write the code, performed the analysis, and wrote the manuscript. The author read and approved the final manuscript.

**Review History**
The review history is available as Additional file 2.

**Funding**
JF thanks the Bio-X Stanford Interdisciplinary Graduate Fellowship for support while writing this manuscript.

## Availability of data and materials

The code supporting the analysis done here is publicly available on GitHub https://github.com/jfukuyama/DeepOrShallow [40] and archived on Zenodo [41]. The data, initially published in [24], is also available at the GitHub repository and on Zenodo. All analyses were done in R [42], and plots were made with ggplot2 [43].

## Ethics approval and consent to participate

Not applicable.

## Consent for publication

Not applicable.

## Competing interests

The author declares that she has no competing interests.

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

## 
