## [Review history. (PDF 95 kb) · Genome Biology]

Review History

First round of review

Reviewer 1

Are you able to assess all statistics in the manuscript, including the appropriateness of statistical tests used?

Yes.

Comments to author:

This is an interesting article that provide some surprising answers about a ubiquitous analysis technique in microbiome studies: namely, analysis of UniFrac distances between samples.

The key result is that there is an implicit weight on the early versus recent ancestral history in weighted and unweighted UniFrac distances. This result is very surprising, and very important. Microbial ecologists typically only include the analysis of UniFrac in their paper if it indicates differences between samples, and so understanding why there are differences in weighted and unweighted UF in some datasets and not others sheds significant light on community differences (the author makes this point very nicely in the Discussion). This is also put into context using other phylogenetically weighted distance metrics. I think this is an important paper with very interesting results and strong justification.

It is a very elegant idea to look at the contribution of each branch to the different metrics, and the figures strongly support the conclusions. The technical results appear very reasonable and the proofs are logically laid out. It is wonderful that the author makes code available to reproduce the figures.

A small number of major concerns

- The introduction should contain slightly more details on the general approach. The author could introduce that they use both mathematical and data analytic evidence to support the claims.
- Similarly, I think the author should spend 2-5 sentences in the introduction providing some intuition for the result that UniFrac distance, which appears (from its definition) to weight all branches equally (i.e., abundance gives the only weight). The results and discussion clarify the intuition, but since the result is counterintuitive I think it should be presented upfront as well.

Addressing some minor concerns would help the readability and clarity of exposition:

- Page 4: a reference is needed for the earth-mover UniFrac interpretation paper (Matsen and Evans? JRSS-B, I think)
- The connection between $\alpha = 0$ as similar to unweighted UniFrac could be strengthened to help readers who are not familiar with generalized unifrac
- "started log transformation" p8 l 8-9. Is this a typographical error? If not, would be great to have a reference or to explicitly state the transformation.
- I think would be great to include some references to papers that use and weighted and unweighted unifrac to reinforce the prevalence of these methods. DPCoA and gUF are newer, so while it would be great to have references that use those techniques as well, I don't think it's as critical if they don't exist yet.
- p14 The interpretation of deep branches having fewer degrees of freedom is very nice. It would be great to have this analogy in the introduction or abstract. I think it makes the results more intuitive for ecologists, which is important because in some ways the results are surprising.

Reviewer 2

Are you able to assess all statistics in the manuscript, including the appropriateness of statistical tests used?

Yes.

Comments to author:

I would like to thank the authors for their well written manuscript. I believe the points raised are important and underappreciated by those using phylogenetic beta-diversity measures. However, I have a few concerns with the conclusions drawn from the provided analyses that I believe need to be addressed.

*** Minor ***

- The 16S rRNA gene often doesn't have the resolution to resolve species, let along strains (e.g. SILVA explicitly doesn't curate species for this reason). This point is not relevant to the current manuscript, but the Introduction gives the impression that the 16S rRNA genes is often used for strain-level community profiling.

- "The opposite problem..." is not clear which problem you are referring to.

- Previous work has compared different phylogenetic beta-diversity measures. How do your results relate to the concept of MRCA, CL, and CT measures given in Parks and Beiko, ISME J, 2013 (PMID 22855211)?

- The scale bars or trees in Figure 2 appear to be incorrect. As shown, it looks like unweighted UniFrac (left panel) has substantial contribution from branches close to the root.

- I believe the idea of "glomming" taxa together is explore to some extent in Stripped UniFrac where there is a mode to ignore all leaf branches in order to reduce computation time by half (PMID 30377368).

- A scale is required for the axes in Figure 6 to be correctly interpreted. Is the unit distance the same for all axes as shown? If not, can the figure be adjusted so this is true.

*** Major ***

- Do the results of Figure 1 hold if you don't normalize by branch length? The results with normalize branch length are only meaningful if deep and shallow branches are generally of the same length. It is unclear that this is generally true for phylogenetic trees.

- Do the results of Figure 1 hold in general are or they specific to the data set evaluated? A simulation study considering different tree topologies and different distributions of taxa between pairs of samples would help establish under what conditions these results hold. I believe this is of concern since it is trivial to build an example where all the distance contributed by weighted UniFrac is due to leaf branches which directly contradicts the conclusions drawn here.

- Page 9, Line 13 it is noted that almost half of the unweighted UniFrac distance is from the leaf branches. It would be helpful to indicate that leaf branches also constitute half the branches in the tree. With this insight, it would be helpful to correlate the proportion of unweighted UniFrac distance with the number of

branches considered in the tree as one moves from the leaves to the root. I suspect this correlation is very high.

- I have trouble connecting the text of the paragraph starting at Page 10, Line 19 with Figure 3. The text indicates that some deep branches can be completely removed from the tree without impacting the unweighted UniFrac measure. I don't believe this is true since all branches contribute to the denominator of unweighted UniFrac. Figure 3 breaks the original tree into a forest of subtrees, but this forest contains all branches in the original tree. If there are branches that do not contribute to the unweighted UniFrac distance can an example be shown where these branches don't appear in the resulting forest of subtrees?

I would like to thank the reviewers for their comments and for the time they spent with the paper. I believe that their comments have improved the manuscript, and I hope they agree.

The line and page numbers referred to in my responses below correspond to line and page numbers in `phyord.pdf` and `phyord-changes-highlighted.pdf`, included as supplemental materials in the submission.

Reviewer 1's major comments

- The introduction should contain slightly more details on the general approach. The author could introduce that they use both mathematical and data analytic evidence to support the claims.

Response: I have added some text to the introduction describing the approach (page 3, lines 16-17) and giving an overview of the major findings (page 3, lines 25-33).

- Similarly, I think the author should spend 2-5 sentences in the introduction providing some intuition for the result that UniFrac distance, which appears (from its definition) to weight all branches equally (i.e., abundance gives the only weight). The results and discussion clarify the intuition, but since the result is counterintuitive I think it should be presented upfront as well.

Response: I have added a paragraph to the end of the introduction that describes some of the mathematical and empirical results (page 3, lines 25-33) and included the degrees of freedom analogy to provide some of the intuition behind the argument (page 3, lines 21-24).

Reviewer 1's minor comments

- Page 4: a reference is needed for the earth-mover Unifrac interpretation paper (Matsen and Evans? JRSS-B, I think)

Response: I believe the reference was there already, but not in the best place — I moved the citation so that it is directly after the statement about the earth-mover's distance, see page 5, lines 13-14.

- The connection between $\alpha = 0$ as similar to unweighted Unifrac could be strengthened to help readers who are not familiar with generalized unifrac.

Response: Thank you for the suggestion. I have added to the section on generalized Unifrac, in particular about the intuition behind the tunable parameter in gUF and what the exact relationship is between gUF with $\alpha = 0$ and unweighted Unifrac, see page 5, lines 24-33 and page 6, lines 1-5.

- "started log transformation" p8 l 8-9. Is this a typographical error? If not, would be great to have a reference or to explicitly state the transformation.

Response: I agree that the transformation should have been written explicitly. The started log transformation is $x \mapsto \log(a + x)$ for some $a > 0$, in this case $a = 1$. The transformation is usually given with Tukey's *Exploratory Data Analysis* as a reference [1], and the transformation has been shown to be approximately variance stabilizing for count data by in Rocke (2003) [2]. I have added both references and written out the transformation in the revised text, see page 9, lines 3-5.

- I think would be great to include some references to papers that use unweighted and weighted unifrac to reinforce the prevalence of these methods. DPCoA and gUF are newer, so while it would be great to have references that use those techniques as well, I don't think it's as critical if they don't exist yet.

Response: This is a good suggestion. I have added references to papers using Unifrac, generalized Unifrac, and DPCoA. I have also added references to some papers that use more than one distance (this is done fairly commonly, usually with weighted and unweighted Unifrac) in an effort to get a fuller picture of the data. See page 3, lines 9-15.

- p14 The interpretation of deep branches having fewer degrees of freedom is very nice. It would be great to have this analogy in the introduction or abstract. I think it makes the results more intuitive for ecologists, which is important because in some ways the results are surprising.

Response: It's good to know that the degrees of freedom analogy is helpful, and I've added it to the introduction page 3 lines 21-24.

Reviewer 2's minor comments

- The 16S rRNA gene often doesn't have the resolution to resolve species, let alone strains (e.g. SILVA explicitly doesn't curate species for this reason). This point is not relevant to the current manuscript, but the Introduction gives the impression that the 16S rRNA genes is often used for strain-level community profiling.

Response: I believe the reason the text gave the impression that 16S was used for strain-level profiling was my use of the word "strain" when I should have used "taxon". I have changed the text in that paragraph of the introduction, page 2, lines 9 and 18.

- "The opposite problem" is not clear which problem you are referring to.

Response: I have changed the text to clarify, see page 2 lines 22-23.

- Previous work has compared different phylogenetic beta-diversity measures. How do your results relate to the concept of MRCA, CL, and CT measures given in Parks and Beiko, ISME J, 2013 (PMID 22855211)?

Response: Parks and Beiko note that phylogenetic measures can be classified as MRCA, CL, or CT, depending on the branches that influence the calculation of the distance. Weighted Unifrac is an MRCA measure, while unweighted Unifrac is a CT measure. This at first seems to be at odds with our results, which show that in practice, unweighted Unifrac depends less on the deep branches than weighted Unifrac. However, I believe this actually

solves something that is a bit puzzling in Parks and Beiko, which is that the categorization of the distances into MRCA/CL/CT doesn't fit well with the empirical clustering of the distances: the CT classification spans the four clusters they find, and the MRCA and CL classification span three of the four clusters.

The results here, both mathematical and empirical, suggest a reason for the lack of alignment: even though unweighted UniFrac technically depends on all of the branches, the form of the distance means that in practice, the deep branches will be less important.

I have added text to the discussion on Parks and Beiko and its relationship to the results in this paper, see page 20 lines 16-33 and page 21 lines 1-4.

- The scale bars or trees in Figure 2 appear to be incorrect. As shown, it looks like unweighted UniFrac (left panel) has substantial contribution from branches close to the root.

Response: Thank you for noticing this. The scale was correct, but it was not described in the legend. The scale represents \log_{10} of the branch contributions, so more strongly negative values correspond to smaller branch contributions. I updated the caption for the figure to describe the scale, see page 11, lines 7-9.

- I believe the idea of "glomming" taxa together is explored to some extent in Stripped UniFrac where there is a mode to ignore all leaf branches in order to reduce computation time by half (PMID 30377368).

Response: Thank you for the reference, I have added a note and reference to the text about this paper and its use of glomming. See page 15, lines 32-33.

- A scale is required for the axes in Figure 6 to be correctly interpreted. Is the unit distance the same for all axes as shown? If not, can the figure be adjusted so this is true.

Response: The fraction of variance explained by each of the axes gives a scale for the axes implicitly. However, I agree that the figure is much easier to read with a scale marked, and I have updated it accordingly, see Figure 7.

Reviewer 2's major comments

- Do the results of Figure 1 hold if you don't normalize by branch length? The results with normalized branch length are only meaningful if deep and shallow branches are generally of the same length. It is unclear that this is generally true for phylogenetic trees.

Response: The results of Figure 1 still hold if you don't normalize by branch length: see the figure below, which is the unnormalized analog of Figure 1. The figure and the code used to create it are available at github.com/jfukuyama/DeepOrShallow, but I haven't included it in the paper for space reasons.

From a conceptual point of view, I think that the contributions normalized by branch length are the correct quantities to look at. This is for essentially the reason the reviewer gives: if there is confounding between branch length and depth in the tree, giving the unnormalized branch contributions gives a combination of information about the contribution of the branch due to its position in the tree and due to its length. We just want to know about how branches in different areas in the tree contribute, and normalizing by branch length takes out the branch length information that we don't want. I have updated the text to clarify why we want to look at contributions per unit branch length and have changed the wording from "branch contributions" to "normalized branch contributions", see page 9, lines 19-27.

That being said, in the trees we have looked at, there hasn't been a strong relationship between branch length and position in the tree, and because of that there is also not much difference between the normalized and unnormalized branch contributions.

- Do the results of Figure 1 hold in general or they specific to the data set evaluated? A simulation study considering different tree topologies and different distributions of taxa between pairs of samples would help establish under what conditions these results hold. I believe this is of concern since it is trivial to build an example where all the distance contributed by weighted UniFrac is due to leaf branches which directly contradicts the conclusions drawn here.

Response: I agree that simulations help establish robustness and intuition, and I particularly like the idea of looking at what happens in a case where all of the distance contributed by weighted Unifrac is contributed by the branches directly above the leaves. I have added three different simulation schemes:

- Simulation 1: the samples come in two clusters with different sets of dominant taxa, which are pulled at random from the tree.
- Simulation 2: the samples come along a gradient, different points along the gradient corresponding to higher or lower relative abundance of a given clade.
- Simulation 3: based on a situation in which the true weighted Unifrac distance places all the weight on the leaves. When we add even a little bit of noise (less than what would be expected for count data), we recover the same pattern.

In all situations we see the same pattern. The closest thing to an exception is the no-noise version of simulation 3, in which weighted Unifrac, unweighted Unifrac, and all the generalized Unifrac distances are due entirely to the leaf branches. I consider this only a partial exception because my contention is that the distances respect an order in which unweighted Unifrac places more weight on the shallow branches than weighted Unifrac, and having both distances place the same weight on the shallow branches is weakly consistent with that.

Simulations 1 and 2 are described and branch accumulation plots shown in the supplemental material. A description of simulation 1 and the results are given on page 11, lines 20-33, page 12, and page 13, lines 1-20, a more detailed description of the simulation setup is on page 28, lines 26-33 and page 29, lines 1-19, and the results are shown in Figure 3.

This comment also suggests that I wasn't clear enough in the manuscript in the point I was trying to make: my argument is not that unweighted Unifrac places a high weight in an absolute sense on the shallow branches (it's not clear to me what that would even mean), but that it consistently places a higher weight on the shallow branches than weighted Unifrac. The math, the branch decompositions on real data, and the branch decompositions on simulated data all support this finding. I have changed the wording in a couple places to make this more clear, see page 1 lines 22 and 23, page 3 line 19, 21, 27 for example.

- Page 9, Line 13 it is noted that almost half of the unweighted UniFrac distance is from the leaf branches. It would be helpful to indicate that leaf branches also constitute half the branches in the tree. With this insight, it would be helpful to correlate the proportion of unweighted UniFrac distance with the number of branches considered in the tree as one moves from the leaves to the root. I suspect this correlation is very high.

Response: This is a good suggestion, and one that I had not considered since I was primarily interested in the difference between the different Unifrac measures and not in the absolute proportion of the distances. I have changed Figure 1 so that the horizontal axis shows the fraction of branches considered instead of the number of descendants, and I believe it makes the figure much more readable. The change to the figure and caption are on page 10, lines 1-9, and the text describing the new plot is on page 9, lines 18-33. Changing the figure also required changing some of the text describing the figure, see page 10, lines 14-25.

The accumulation curve for unweighted Unifrac does not exactly follow the line representing equality between the fraction of the distance and the fraction of the branches considered: the shallow branches represent less than half, and the deep branches are proportionally less represented than they would be with equality. We still see a dramatic difference between unweighted and weighted unifrac in terms of the fraction of weight put on the deep branches.

- I have trouble connecting the text of the paragraph starting at Page 10, Line 19 with Figure 3. The text indicates that some deep branches can be completely removed from the tree without impacting the unweighted UniFrac measure. I don't believe this is true since all branches contribute to the denominator of unweighted UniFrac. Figure 3 breaks the original tree into a forest of subtrees, but this forest contains all branches in the original tree. If there are branches that do not contribute to the unweighted UniFrac distance can an example be

shown where these branches don't appear in the resulting forest of subtrees?

Response: Thank you for catching this. The figure is correct but the text describing it is not. The mathematical result is that the tree can be broken at certain nodes, creating a forest, without changing the unweighted Unifrac distance remain unchanged. Each of the original branches remains in the resulting forest, but the ancestry relationships between nodes and leaves changes. If it were not for the normalization by total branch length in the unweighted Unifrac distance, you could remove branches in addition to removing connections.

I have changed the text to reflect this, page 13 lines 25-30 and page 14 line 32.

References

- [1] Tukey, J.W.: Exploratory Data Analysis. Addison-Wesley, ??? (1977)
- [2] Rocke, D.M., Durbin, B.: Approximate variance-stabilizing transformations for gene-expression microarray data. *Bioinformatics* **19**(8), 966–972 (2003)